# The Association of Immune-Related Adverse Events with the Efficacy of Atezolizumab in Previously Treated Advanced Non-Small-Cell Lung Cancer Patients: A Single-Center Experience

**DOI:** 10.3390/cancers16172995

**Published:** 2024-08-28

**Authors:** Filip Marković, Mihailo Stjepanović, Natalija Samardžić, Milica Kontić

**Affiliations:** 1Clinic for Pulmonology, University Clinical Center of Serbia, 11000 Belgrade, Serbia; flp.mark@gmail.com (F.M.); natalija.samardzic@kcs.ac.rs (N.S.); milica.kontic@med.bg.ac.rs (M.K.); 2Faculty of Medicine, University of Belgrade, 11000 Belgrade, Serbia

**Keywords:** immunotherapy, NSCLC, immune-related adverse events, atezolizumab, checkpoint inhibitors

## Abstract

**Simple Summary:**

This study explores the association of immune-related adverse events during atezolizumab therapy with better outcomes for patients with advanced non-small-cell lung cancer who had already received chemotherapy in previous lines of treatment. This study found that patients who experienced an immune-related adverse event while undergoing atezolizumab treatment had longer median progression-free survival compared to those who did not. Specifically, patients that experienced side effects had a median progression-free survival of 13.03 months, compared to just 3.4 months for those without side effects. Additionally, these side effects were more common in patients who were in better overall health when they started the treatment. The findings suggest that immune-related adverse events could be predictive in this patient population.

**Abstract:**

Immune checkpoint inhibitors (ICIs) are pivotal in managing metastatic non-oncogene addicted non-small-cell lung cancer (NSCLC). They have unique toxicities known as immune-related adverse events (irAEs). Previous studies have linked irAEs during atezolizumab-based first-line treatments in advanced NSCLC with improved outcomes. This study explored the association between irAEs and the efficacy of atezolizumab in advanced NSCLC patients who had previously received platinum-based chemotherapy. The study involved 105 advanced NSCLC patients who received atezolizumab monotherapy after progressing on at least one line of platinum-based chemotherapy from a single academic institution in Serbia. Data were obtained from a hospital lung cancer registry. Among the participants, 63.8% were male, with the majority being current (53.3%) or former smokers (37.1%). About half had a good performance status (ECOG PS 0–1) at the start of atezolizumab treatment. irAEs occurred in 23 patients (21.9%). The median progression-free survival (mPFS) was significantly longer for patients with irAEs (13.03 months) compared to those without (3.4 months) (HR 0.365 [95% CI, 0.195–0.681], *p* = 0.002). irAEs and ECOG PS 0–1 were predictors of longer mPFS, with irAEs being more common in patients with good performance status (*p* = 0.01). irAEs were linked to improved mPFS in NSCLC patients treated with atezolizumab after multiple lines of platinum-based chemotherapy.

## 1. Introduction

Lung cancer continues to be the leading cause of cancer-related deaths globally [1]. Non-small-cell lung cancer (NSCLC) represents 80–85% of all lung cancer cases, with most patients being diagnosed at advanced stages, making the disease incurable [2,3]. However, over the past decade, there has been an improvement in 5-year survival rates among patients with advanced NSCLC, mainly due to development of new therapeutic modalities, including immune checkpoint inhibitors (ICIs) [4,5,6]. Nowadays, ICIs form the foundation for the treatment of metastatic non-oncogene addicted NSCLC [7]. ICIs display a unique toxicity profile, and their side effects are named immune-related adverse events (irAE). Although their exact underlying mechanism of onset is unclear, irAEs can affect any organ system [8]. Mild to moderate grades of irAEs are usually easily manageable by corticosteroids or substitution therapy, while severe irAEs could be fatal in some instances [9]. Interestingly, the manifestation of irAEs in advanced NSCLC patients undergoing ICI therapies, including those on first-line atezolizumab regimens, has been consistently associated with improved objective response rates, progression-free survival, and overall survival across multiple clinical trials [10].

Patients with mild to moderate grades of irAEs have derived the greatest clinical benefit [10]. The potential predictive capabilities of irAE occurrence as a biomarker in NSCLC patients undergoing atezolizumab therapy deserve further investigation. Atezolizumab was also approved for patients with locally advanced or metastatic NSCLC following platinum-based chemotherapy on the basis of the results of the phase III OAK trial [11]. Since then, there have been slight indications of favorable outcomes with anti-PD-L1 inhibitors such as atezolizumab compared to anti-PD-1 inhibitors in pretreated advanced NSCLC patients [12]. Due to rigorous eligibility criteria, the patient populations enrolled in large-scale clinical trials do not necessarily resemble that which is encountered in every day clinical practice. Patients that have been heavily pretreated, as well as those with a poor Eastern Cooperative Oncology Group performance status (ECOG PS), untreated brain metastases, and specific comorbidities, are usually excluded from such trials [11]. We find it significant to determine whether findings regarding the association between irAEs and atezolizumab treatment efficacy could be reproduced in a real-world setting. 

Due to factors such as healthcare policies and the availability or absence of insurance coverage, there is uneven global access to ICIs, including atezolizumab, particularly in developing nations. Our patient cohort represents a real-world scenario where atezolizumab is often administered as a later-line treatment, unlike in more affluent regions.

The aim of this study is to determine the association of irAEs with atezolizumzab efficacy in NSCLC patients that have previously received multiple lines of platinum-based chemotherapy.

## 2. Materials and Methods

### 2.1. Patients

This study included 105 patients with advanced NSCLC that had underwent treatment with atezolizumab monotherapy following disease progression on at least one prior line of platinum-based chemotherapy at a single academic institution in Serbia. In 2019, atezolizumab monotherapy became available for the treatment of advanced NSCLC patients whose disease has previously progressed on platinum-based chemotherapy. All patients underwent PD-L1 testing. Also, testing for EGFR mutations by cobas^®^ EGFR Mutation Test v2 and ALK rearrangement by immunohistochemistry was carried out in this patient population prior to initiating first-line treatment. None of the included patients had a known driver oncogene. Testing for other molecular alterations was not part of routine practice in Serbia at the time this study’s conduction.

All patients were routinely treated and monitored at the Clinic for Pulmonology, University Clinical Centre of Serbia. Clinicians were encouraged to report and grade the observed irAEs according to CTCAE v5 [13]. Objective response to atezolizumab therapy was reported according to iRECIST criteria [14].

### 2.2. Data

Data were retrieved from a hospital-based lung cancer registry that prospectively collects the demographical, clinical, pathological, and molecular characteristics, as well as the treatment and survival data, of the patients diagnosed and treated at our center. All data were collected anonymously. Retrospective analysis of the gathered data was carried out without the need for ethical approval. The study was performed in accordance with the Declaration of Helsinki. 

### 2.3. Statistical Analysis

Descriptive methods were used on demographic characteristics of patients. Baseline information is presented as number of patients and percentages. Median progression-free survival (PFS) and overall survival (OS) were calculated as time from start of therapy until therapy discontinuation or date of death. Patients still alive on the last day of follow-up were censored. 

Median PFS and OS were estimated by the Kaplan–Meier method and compared by the log-rank test. Univariable and multivariable Cox proportional hazard regression models were used to calculate hazard ratios (HRs) and confidence intervals (CIs). In the univariate analysis, covariates included age (<70 years vs. ≥70 years), sex, histology (non-squamous vs. squamous cell lung cancer), smoking status (current vs. former smoker or never smoker), PD-L1 expression (≥50% vs. <50% or unknown), ECOG PS (0/1 vs. ≥2), and the presence of irAEs (yes vs. no). Multivariate analysis included variables with a significance level of *p* < 0.10 in the univariate analysis. The Chi-square test was used to determine the association between response to treatment and the presence of irAEs. The calculated *p*-values are two-sided. We used SPSS v26 for statistical analysis.

## 3. Results

### 3.1. Patient Characteristics and irAE Profiles

There were 105 advanced NSCLC patients that started treatment with atezolizumab and were previously treated with chemotherapy. Among them, the mean age was 62.31 (35–82), and 63.8% were male. Most participants were smokers or former smokers, with these categories accounting for 53% and 39% of the cohort, respectively. Adenocarcinomas were detected in 56% of cases, and squamous cell carcinomas were detected in 41% of cases. Patients received 10.1 cycles of chemotherapy (2–26) on average prior to atezolizumab treatment (Table 1).

The median PFS was 4.67 months (95% CI, 1.3–8.02 months). The median OS was 35.8 months (95% CI, 27.8–43.7 months). In terms of the best overall responses to atezolizumab, we noted complete response, partial response, and stable disease, with percentages of 1%, 17.1%, and 36.2%, respectively, adding up to overall response rate (ORR) of 18.1% and disease control rate (DCR) of 54.3% (Table 1).

Overall, 23/105 (21.9%) patients experienced an irAE. irAEs were more likely to occur in female patients and among those with ECOG PS 1–2 (Table 2).

irAEs of grade 1–2 were registered in 19/105 (18.1%) patients, and 4/105 patients (3.8%) had irAEs of grades 3–4, as per CTCAE v5 criteria (Table 3). Atezolizumab therapy was permanently discontinued in three patients with grade 3–4 hepatotoxicity as per the ESMO Guidelines for the management of toxicities from immunotherapy [15]. Systemic corticosteroid therapy was used in patients that developed pneumonitis, as all registered cases were of grade 2 per CTCAE v5. All of the patients that developed hepatotoxicity were treated with systemic corticosteroids on account of the irAE grade, as were the patients with fatigue as, in all cases, symptoms persisted despite rest and restricted activities of daily living.

### 3.2. Association between irAEs and Atezolizumab Efficacy 

With regard to response to treatment, the occurrence of irAEs was predictive for better responses to treatment (*p* = 0.001) (Table 4).

Among patients that experienced irAEs, the median PFS was 13.03 months (95% CI, 6.98–19.08 months) vs. 3.4 months (95% CI, 1.73–5.06 months) in those without irAEs. Hence, the development of irAEs was significantly associated with improved mPFS (HR 0.365 [95% CI, 0.195–0.681] *p* = 0.002) (Figure 1).

Although there was substantial numerical difference in median OS between the patients without irAE occurrence—34.03 months (95% CI, 25.67–42.39 months) vs. 51.36 months (95% CI, 23.11–79.62 months)—in those with irAEs, a statistically significant difference was not achieved (HR 0.639 [95% CI, 0.354–1.152] *p* = 0.136). 

In our multivariable Cox regression analysis, besides the occurrence of irAEs, ECOG PS 0–1 at the time of atezolizumab therapy onset was also predictive for longer PFS. On the other hand, sex, age (<70 years vs. ≥70 years), PD-L1 expression, histology, and smoking status were not predictive of improved PFS (Table 5).

## 4. Discussion

Our results have shown that irAEs were associated with atezolizumab efficacy in terms of prolonged mPFS and ORR among pretreated patients with advanced NSCLC. The association of irAEs with atezolizumab efficacy was further supported by our multivariable analysis. 

In our cohort, irAEs were observed in 21.9% of patients. In large-scale trials of atezolizumab therapy following progression on platinum-based chemotherapy, TAIL and OAK, irAEs were found in 9.6% and 33.2% of patients, respectively [16,17]. These differences could be a reflection of the enrollment criteria, as the OAK trial did not include patients with specific comorbidities, brain metastases, more than one previous line of chemotherapy, and those with ECOG PS ≥ 2, unlike TAIL and our study. 

A meta-analysis conducted by Sun X et al. explored the occurrence and types of irAEs among NSCLC patients undergoing PD-1 and PD-L1 inhibitor therapy. Their analysis included 16 studies, and among them, there were those in which ICI therapy, including atezolizumab, was utilized in later lines of treatment following progression on chemotherapy. Both the overall incidence of irAEs (22%) and the incidence of high-grade irAEs (4%) were almost identical to those of our study (Table 1) [18]. However, no association between efficacy of treatment and occurrence of irAEs was established.

In the OAK trial, mOS and mPFS was found to be longer, in addition to ORR being higher, in patients with irAEs than in those without [10,19]. More recent publications investigating the relationship between occurrence of irAEs and treatment outcomes of advanced NSCL patients have reported similar findings [20,21,22]. These studies included both previously treated patients and those receiving ICI-based combination regimens. 

Our study did not show statistically significant benefits in terms of mOS. The administration of multiple cycles of chemotherapy before initiating atezolizumab (mean of 10.1 cycles (2–21)) might have reduced the impact of irAEs on mOS. Nevertheless, the significant prolongation of mPFS and higher ORR values among those who experienced irAEs underscore the treatment’s effectiveness in this specific subgroup of patients. 

Of the 23 patients who experienced an irAE, the majority (82.6%) were classified as grade 1–2 as per CTCAE v5. It could be speculated that the efficacy of atezolizumab observed in our cohort may have been influenced by this subset of patients. This is in alignment with the conclusions of Socinski et al. that it is the patients who experience low-grade irAEs that stand to benefit the most from atezolizumab-based regimens for NSCLC [10]. These authors argued that this may be due to the life-threatening nature of high-grade irAEs, warranting treatment discontinuation and systemic immunosuppression, which may hinder ICI efficacy. Such a statement may be supported by our results, as the minority of the patients that had an irAE either discontinued ICI treatment (13.1%) or needed the systemic use of immunosuppressants (43.4%). In contrast to the publication by Socinski et al., which focused on atezolizumab-based regimens used as first-line treatment in large-scale trials, our study focused on patients who had received atezolizumab therapy following progression on multiple lines of platinum-based chemotherapy. Additionally, we also included patients with squamous NSCLC, a subgroup not addressed in Socinski et al.’s analysis.

Other authors have also found an association between better clinical outcomes and low-grade irAEs in NSCLC patients undergoing ICI therapy. In a retrospective multicenter study by Wang et al., as well as a meta-analysis by Zhang et al., the authors found that those with low-grade irAEs derive the greatest clinical benefit while undergoing ICI monotherapy [22,23]. Both publications included patients that received previous lines of chemotherapy. Additionally, in a subgroup analysis, Zhang et al. found that the development of skin, endocrine, and gastrointestinal irAEs is related to better survival outcomes. In a more recent meta-analysis by Lin et al. that also included ICI-based combination therapy for advanced NSLC, it was also evident that these types of irAEs are associated with favorable outcomes [20]. Our sample of patients is too limited in size to draw definite conclusions; however, skin, endocrine, and gastrointestinal irAEs did occur in 56.5% of our patients that experienced irAEs. While all of them were of grades 1–2, occurrence of these specific types of irAEs could have impacted the favorable outcomes in this patient subset.

Nearly half of our patients had ECOG PS ≥ 2 (49.5%) prior to atezolizumab initiation. Despite being underrepresented in large-scale clinical ICI trials, some reports show that patients with poor ECOG PS account for up to 40% of patients diagnosed with advanced non-small-cell lung cancer (NSCLC) who are scheduled to undergo this treatment [24]. A higher proportion of patients with poor performance status were observed in our study primarily due to the fact that all of them received atezolizumab as a second-line therapy or in subsequent lines of therapy. This aligns with the results reported by Meyers et al. as, in their study, advanced NSCLC patients with an ECOG PS > 2 were statistically more common among those receiving ICI therapy in the second line or later rather than as a first-line treatment [24]. In the multivariable analysis, good ECOG PS was also a predictive factor for longer PFS. Several trials conducted in real-world settings assessing ICI therapy in patients with NSCLC have similarly indicated that poor ECOG PS correlates with worse clinical outcomes [24,25,26].

We found that irAEs were more likely to occur in patients with ECOG PS 0–1 and among female patients. 

In our study, female patients were more likely to develop irAEs. However, several real-world studies investigating risk factors for the occurence of irAEs in NSCLC patients have found no association of sex with irAE development [27,28,29]. All of the studies included advanced NSCLC patients receiving an ICI in the second line and further lines. In a meta-analysis of patients with multiple cancer types, including NSCLC, undergoing ICI therapy by Jing et al., no statistically significant difference regarding irAE development between the sexes was observed [30].

Good ECOG PS has been associated with the development of irAEs in several real-world data-based studies involving NSCLC patients undergoing ICI therapy [21,31]. It seems that patients of good general condition are more prone to developing irAEs, suggesting that they should be more closely followed up with. Close monitoring could be crucial, as early recognition of irAEs and their timely intervention could eliminate the need for prolonged periods of immunosuppressant use, ICI interruption, or discontinuation, which can be detrimental [32,33]. Such proactive monitoring may be crucial in optimizing treatment outcomes and ensuring the well-being of patients undergoing ICI therapy.

## 5. Limitations 

Atezolizumab monotherapy became available for treating NSCLC patients post-platinum-based chemotherapy progression at our center in 2019. Patients undergoing platinum-based chemotherapy in any line at the time were considered eligible to receive atezolizumab upon evidence of disease progression and the clinician’s approval. This should explain the fact that 38.1 % of the enrolled patients received at least three lines of platinum-based chemotherapy prior to atezolizumab. The lack of data regarding the PD-L1 status for 21% of the patients could be contributed to the fact that they were diagnosed with advanced NSCLC prior to the introduction of first-line pembrolizumab therapy for the PD-L1 > 50% subset of patients, and thus, PD-L1 status was not routinely tested for. As the use of atezolizumab following progression of platinum-based chemotherapy is not contingent on PD-L1 status, PD-L1 was not routinely tested for either from the archival sample or the one obtained post-platinum-based chemotherapy progression. At the time of conducting this study, the patients were only tested for oncogenic alterations in ALK and EGFR genes. More comprehensive genomic testing would have allowed for the identification of other oncogenic alterations such as HER2, ROS1, RET, and NTRK, as well as STK11 and KEAP1 mutations that have been associated with poor outcomes to ICI therapy [34]. Additionally, this would have allowed for the identification of certain genetic profiles, such as KRAS/TP53 co-mutants that have been associated with favorable response to ICIs [34,35]. Taken together, this could help better inform patient stratification and help tailor treatment strategies more precisely [36]. Broader gene panel testing could have identified specific HLA profiles and specific single-nucleotide polymorphisms that have been recently associated with the occurrence of irAEs [37]

## 6. Conclusions

Occurrence of irAEs was associated with prolonged mPFS and better ORR in NSCLC patients treated with atezolizumab that have previously received multiple lines of platinum-based chemotherapy in our group of patients. irAEs were more likely to occur in patients of good performance status. 

## Figures and Tables

**Figure 1 cancers-16-02995-f001:**
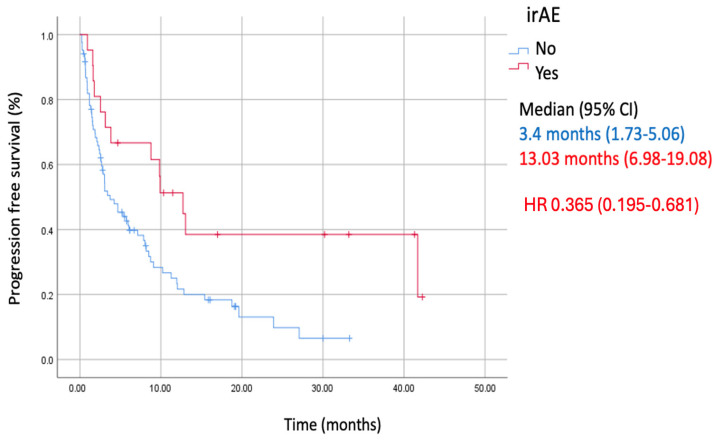
Kaplan–Meier curve of progression-free survival (PFS) in patients with and without immune-related adverse events (irAE).

**Table 1 cancers-16-02995-t001:** Demographic data of included patients.

*N* = 105	*N* (%)
Mean age at treatment start (range) [years]	62.31 (35–82)
Sex	
Male	67 (63.8)
Female	38 (36.2)
Smoking status	
Current	56 (53.3)
Ex-smoker	39 (37.1)
Non-smoker	10 (9.5)
ECOG PS	
0–1	53 (50.5)
≥2	52 (49.5)
Histological diagnosis	
Adenocarcinoma	59 (56.2)
Squamous cell carcinoma	43 (41.9)
Other (NOS)	2 (1.9)
*PD-L1 status	
50–100%	13 (12.4)
1–49%	27 (25.7)
<1%	43 (41)
No data available	22 (21)
Stage at the time of diagnosis	
IIIB	16 (15.2)
IV	89 (84.8)
Atezolizumab line of treatment	
II	19 (18.1)
III	21 (20)
IV	46 (43.8)
V	18 (17.1)
VI	1 (1)
Mean number of chemotherapy cycles (range)	10.01 (2–26)
Immune-related adverse events	23 (21.9)
Grade 1–2	19 (18.1)
Grade 3–4	4 (3.8)
No immune-related adverse events	82 (78.1)
Best response to atezolizumab (iRECIST)	
^α^PD	48 (45.7)
^β^SD	38 (36.2)
^δ^PR	18 (17.1)
^μ^CR	1 (1)
Real world °DCR	54.3%
Real world ^ö^ORR	18.1%
Median PFS (95% confidence interval) [months]	4.67 (1.3–8.02)
Median overall survival (95% confidence interval) [months]	35.8 (27.8–43.7)

*PD-L1—programmed death ligand 1; ^α^PD—progressive disease; ^β^SD—stable disease ^δ^PR—partial response; ^μ^CR—complete response; °DCR—disease control rate; ^ö^ORR—overall response rate.

**Table 2 cancers-16-02995-t002:** Characteristics of patients with and without registered irAEs.

	Patients with irAE (*n* = 23)	Patients without irAE (*n* = 82)	*p* Value
**Age**			0.50
<70 years	18 (78.3%)	69 (84.1%)	
≥70 years	5 (21.7%)	13 (15.9%)	
**Sex**			0.02
Male	10 (43.5%)	57 (69.5%)	
Female	13 (56.5%)	25 (30.5%)	
**Histology**			0.43
Non-squamous	15 (61.9%)	46 (56.1%)	
Squamous	8 (38.1%)	36 (43.8%)	
**Smoking status**			0.55
Current	12 (52.2%)	37 (45.1%)	
Former or never smoker	11 (47.2%)	45 (54.9%)	
**PD-L1 status**			0.91
≥50%	3 (13%)	10 (12.2%)	
<50% or unknown	20 (87%)	72 (87.8%)	
**ECOG PS**			0.01
0–1	17 (73.9%)	36 (43.8%)	
2	6 (26.1%)	46 (56.2%)	
**Line of treatment**			0.27
Second and third	11 (47.2%)	29 (35.4%)	
Later	12 (52.2%)	53 (64.6%)	

**Table 3 cancers-16-02995-t003:** Types and grades of irAEs as per CTCAEv5.

irAE Type	Grade 1–2	Grade 3–4
Diarrhea	4 (17.4%)	
Thyroid toxicities	4 (17.4%)	
Skin toxicity	5 (21.7%)	
Pneumonitis	3 (13%)	
Hepatotoxicity	1 (4.3%)	3 (13%)
General (fatigue)	2 (8.6%)	1 (4.3%)

**Table 4 cancers-16-02995-t004:** Response to treatment among patients that have experienced an immune-related adverse event (Yes column) and those that have not (No column).

	Immune-Related Adverse Events	
	Yes—*N* (%)	No—*N* (%)
CR	1 (4.34%)	0 (0)
PR	7 (30.43%)	11 (13.41%)
SD	12 (52.17%)	27 (32.92%)
PD	3 (13.04%)	45 (54.87%)

**Table 5 cancers-16-02995-t005:** Univariable and multivariable regression analysis.

	Univariable Regression Analysis			Multivariable Regression Analysis		
	HR	95% CI	*p*	HR	95% CI	*p*
Age (<70 years vs. ≥70 years)	0.991	0.562–1.748	0.974			
Sex (male vs. female)	1.083	0.676–1.735	0.741			
Histology (non-squamous vs. squamous)	1.056	0.674–1.653	0.813			
Smoking status (current vs. former or never smoker)	1.312	0.837–2.057	0.237			
PD-L1 (≥50% vs. <50% or unknown)	0.917	0.470–1.787	0.798			
ECOG PS (0–1 vs. 2)	7.744	4.440–13.508	<0.001	7.124	4.061–12.497	<0.001
irAE (yes vs. no)	0.365	0.195–0.681	0.002	0.457	0.242–0.863	0.016

## Data Availability

The data presented in this study are available on request from the corresponding author.

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
