# Peer review of "The Association of Immune-Related Adverse Events with the Efficacy of Atezolizumab in Previously Treated Advanced Non-Small-Cell Lung Cancer Patients: A Single-Center Experience"

_cancers, 2024, doi:10.3390/cancers16172995_

Round 1
Reviewer 1 Report
Comments and Suggestions for Authors
This manuscript reports the authors’ study of the association of immune-related adverse events with efficacy of Atezolizumab in previously treated advanced non-small cell lung cancer. The authors examined 105 advanced NSCLC patients who received atezolizumab monotherapy after progressing on at least one line of platinum-based chemotherapy, from 29 a single academic institution in Serbia. They found that patients who experienced immune-related adverse event while undergoing atezolizumab treatment had longer median progression free survival compared to those who did not. Patients that experienced side effects had median progression-free survival of 13.03 months, compared to just 3.4 months for those without side effects. These side effects were more common in patients who were in better overall health when they started the treatment. The authors concluded that occurence of irAEs was associated with prolonged mPFS and better ORR in NSCLC patients treated with atezolizumab that have previously received multiple lines of platinum-based chemotherapy in their group of patients.
These results are interesting, although the size of the patient group is small. Table 4 is not well illustrated. It is not clear what the values in the two columns represent.
Comments on the Quality of English LanguageGenerally acceptable
Author Response
Thank you for taking the time out of your day to deliver constructive feedback on our manuscript. We appreciate your positive remarks regarding the results and have taken your suggestions into consideration.
Regarding Table 4, we have made revisions to improve its clarity and presentation. Specifically, the table now clearly indicates that the values in the two columns represent the differences in responses to therapy as per iRECIST criteria between patients with immune-related adverse events (irAEs) and those without irAEs. We hope that these changes have addressed the issues you raised and that the information is now more comprehensible.
Please see the attachment.

Reviewer 2 Report
Comments and Suggestions for Authors
In this manuscript, the authors evaluated the side effects of atezolizumab in NSCLC cases. This is an interesting study, but some potential points can improve the current version;
1) There are a wide range of lung cancer treatments. It is strongly recommended that the authors use the related publications and conclude why they selected Atezolizumab such as Jianwei Zhu, et al - 2021, Monireh Mohsenzadegan, et al - 2020, Alfredo Tartarone, et al - 2019 and so on..
2) There are some reports about the side effects of ICIs in NSCLC. The authors should compare their results with them, such as Xiaoying Sun, et al - 2019, Fausto Petrelli, et al - 2021 and so on
3) The role of genetic background needs to be addressed in the discussion. Since there is no data regarding ICIs therapy in some populations, the authors should discuss it using related publications such as Zahra Fathi, et al - 2018 and so on...
Comments on the Quality of English Languageminor English error
Author Response
Thank you for taking the time out of your day to deliver your constructive feedback on our manuscript. We appreciate your remarks and have taken your suggestions into consideration.
Comment 1: There are a wide range of lung cancer treatments. It is strongly recommended that the authors use the related publications and conclude why they selected Atezolizumab such as Jianwei Zhu, et al - 2021, Monireh Mohsenzadegan, et al - 2020, Alfredo Tartarone, et al - 2019 and so on..
Response 1: Thank you for mentioning this. We have made the following adjustments in the introduction section according to your suggestions - Row(s) 44-49; 63-66; and 73-76
Comment 2: There are some reports about the side effects of ICIs in NSCLC. The authors should compare their results with them, such as Xiaoying Sun, et al - 2019, Fausto Petrelli, et al - 2021 and so on
Response 2: Thank you for highlighting this. We have compared our findings with large scale radomized control trials involving patients that have received atezolizumab in later lines of therapy (OAK, TAIL). Moreover we also comented on the differences regarding the influence of irAE on treatment outcomes of NSCLC patients in more recent publications (Lin et al. 2023;Cook et al 2024;Wang et al. 2022) as well as meta-analyses by Socinski et al 2023 and Zhang et al 2022.
However we thank you for bringing these publications to our attention. We have since made adjustments in our manuscript.
Row(s): 177-183
Comment 3: The role of genetic background needs to be addressed in the discussion. Since there is no data regarding ICIs therapy in some populations, the authors should discuss it using related publications such as Zahra Fathi, et al - 2018 and so on
Response 3: Once again than you for bringing this issue up. We have made adjustments in the manuscript. Row(s) - 263-272
Please see the attachment.